# Pressureless glass crystallization of transparent yttrium aluminum garnet-based nanoceramics

Xiaoguang Ma[1,2], Xiaoyu Li[1], Jianqiang Li [1,3], Cécile Genevois[4], Bingqian Ma[1], Auriane Etienne[5], Chunlei Wan[6], Emmanuel Véron[4], Zhijian Peng[2] & Mathieu Allix[4]

Transparent crystalline yttrium aluminum garnet (YAG; $Y_3Al_5O_{12}$) is a dominant host material used in phosphors, scintillators, and solid state lasers. However, YAG single crystals and transparent ceramics face several technological limitations including complex, time-consuming, and costly synthetic approaches. Here we report facile elaboration of transparent YAG-based ceramics by pressureless nano-crystallization of $Y_2O_3$–$Al_2O_3$ bulk glasses. The resulting ceramics present a nanostructuration composed of YAG nanocrystals (77 wt%) separated by small $Al_2O_3$ crystalline domains (23 wt%). The hardness of these YAG-$Al_2O_3$ nanoceramics is 10% higher than that of YAG single crystals. When doped by $Ce^{3+}$, the YAG-$Al_2O_3$ ceramics show a 87.5% quantum efficiency. The combination of these mechanical and optical properties, coupled with their simple, economical, and innovative preparation method, could drive the development of technologically relevant materials with potential applications in wide optical fields such as scintillators, lenses, gem stones, and phosphor converters in high-power white-light LED and laser diode.

[1] National Engineering Laboratory for Hydrometallurgical Cleaner Production Technology, CAS Key Laboratory of Green Process and Engineering, Institute of Process Engineering, Chinese Academy of Sciences, 100190 Beijing, People's Republic of China. [2] School of Engineering and Technology, China University of Geosciences, 100083 Beijing, People's Republic of China. [3] School of Chemical Engineering, University of Chinese Academy of Sciences, 100049 Beijing, People's Republic of China. [4] CNRS, CEMHTI UPR 3079, Univ. Orléans, 45071 Orléans, France. [5] Groupe de Physique des Matériaux, UNIROUEN, CNRS, INSA Rouen, Normandie Univ, 76000 Rouen, France. [6] State Key Lab of New Ceramics and Fine Processing, School of Materials Science and Engineering, Tsinghua University, 100084 Beijing, People's Republic of China. These authors contributed equally: Xiaoguang Ma, Xiaoyu Li. Correspondence and requests for materials should be addressed to J.L. (email: jqli@ipe.ac.cn) or to M.A. (email: allix@cnrs-orleans.fr)

Transparent crystalline yttrium aluminum garnet (YAG; $Y_3Al_5O_{12}$) is particularly noteworthy due to its importance as a host material in solid state lasers[1–4], phosphors[5–8], and scintillators[9]. Commercial YAG materials are usually single crystals, which are grown by directional solidification from melts and demonstrate outstanding optical performances[10]. However, such single crystals are limited in size and shape, maximal chemical doping level, and crystal growth rate which imply high production costs. Transparent YAG ceramics have recently proved their ability to compete with these single crystals in domains including optics, electronics, and scintillating devices[1,11–13]. Transparency in these polycrystalline materials is ensured by the absence of light scattering sites (pores, secondary phases) to avoid energy dissipation within the material[13]. Compared to single-crystal technology, transparent ceramics attract significant attention due to geometric versatility, relatively swift scalable manufacturing, and doping flexibility[1,13–15]. Diverse sintering synthetic approaches have been employed for their elaboration, including vacuum sintering, hot isostatic pressing or spark plasma sintering with specific nanometer-scale raw powders[1]. There has been only very few reports of highly transparent nanocrystalline ceramic until the recent work on silicate garnets fabricated by direct conversion from bulk glass starting material in mutianvil high-pressure apparatus[16]. Nevertheless, all these processes require high-pressure and high-temperature sintering conditions, and it remains challenging to reach industrial production due to complex processes and reproducibility problems[16–18]. Ikesue et al.[19] reported the possibility to prepare transparent YAG ceramics by pressureless slip casting and vacuum sintering at about 1800 °C but with large grain size (40–60 μm). However, even though fully densified ceramics with nanometer-scale grain sizes are promised to unprecedented optical, mechanical and electrical properties with applications in lasers, phosphors, and electrical devices[1,13,20,21], pressureless fabrication of transparent nanoceramics has never been reported up to date.

To overcome the drawbacks of both single crystals and powder sintered ceramics, full crystallization from glass is regarded as a

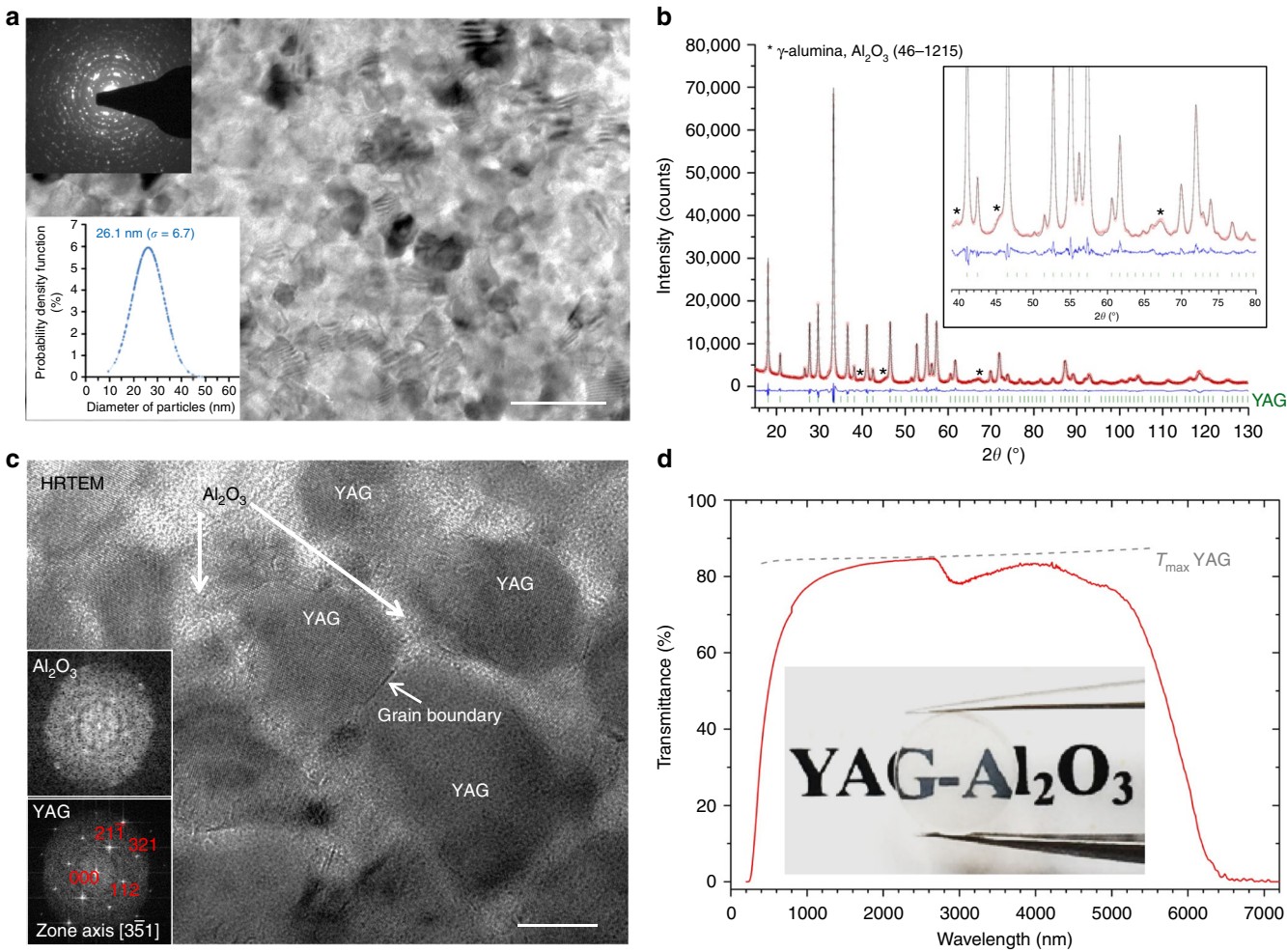

**Fig. 1** Transparent YAG-$Al_2O_3$ composite nanoceramics. **a** Bright field TEM micrograph showing the nanometer-scale microstructure of the fully crystallized material (scale bar corresponds to 100 nm). The selected area electron diffraction pattern shows the crystallinity of the sample with both YAG and $Al_2O_3$ reflections. The size distribution with standard deviation ($\sigma$) of the YAG crystals is also embedded. **b** X-ray diffraction of the YAG-$Al_2O_3$ ceramic crystallized from a 26 $Y_2O_3$–74 $Al_2O_3$ glass via a simple 2 h heat treatment at 1100 °C in air. The Rietveld refinement leads to a 79±2 wt% content of YAG (21±2 wt% of γ-$Al_2O_3$). **c** HRTEM micrograph of the YAG-$Al_2O_3$ ceramic showing the presence of two crystalline phases: thin $Al_2O_3$ areas surrounding YAG nanograins sharing grain boundaries (scale bar corresponds to 20 nm). The related FFT patterns are embedded. **d** Transmission spectra in UV–VIS–NIR and MIR region of the YAG-$Al_2O_3$ ceramic measured through a 1.5 mm thick sample. The dashed gray line corresponds to the theoretical maximum transmission of YAG[53] (the refractive index of YAG single crystal and YAG-$Al_2O_3$ ceramic at 589 nm is 1.82 and 1.77, respectively). The photograph of the YAG-$Al_2O_3$ ceramic sample (diameter = 4.5 mm) placed 2 cm above the text is embedded

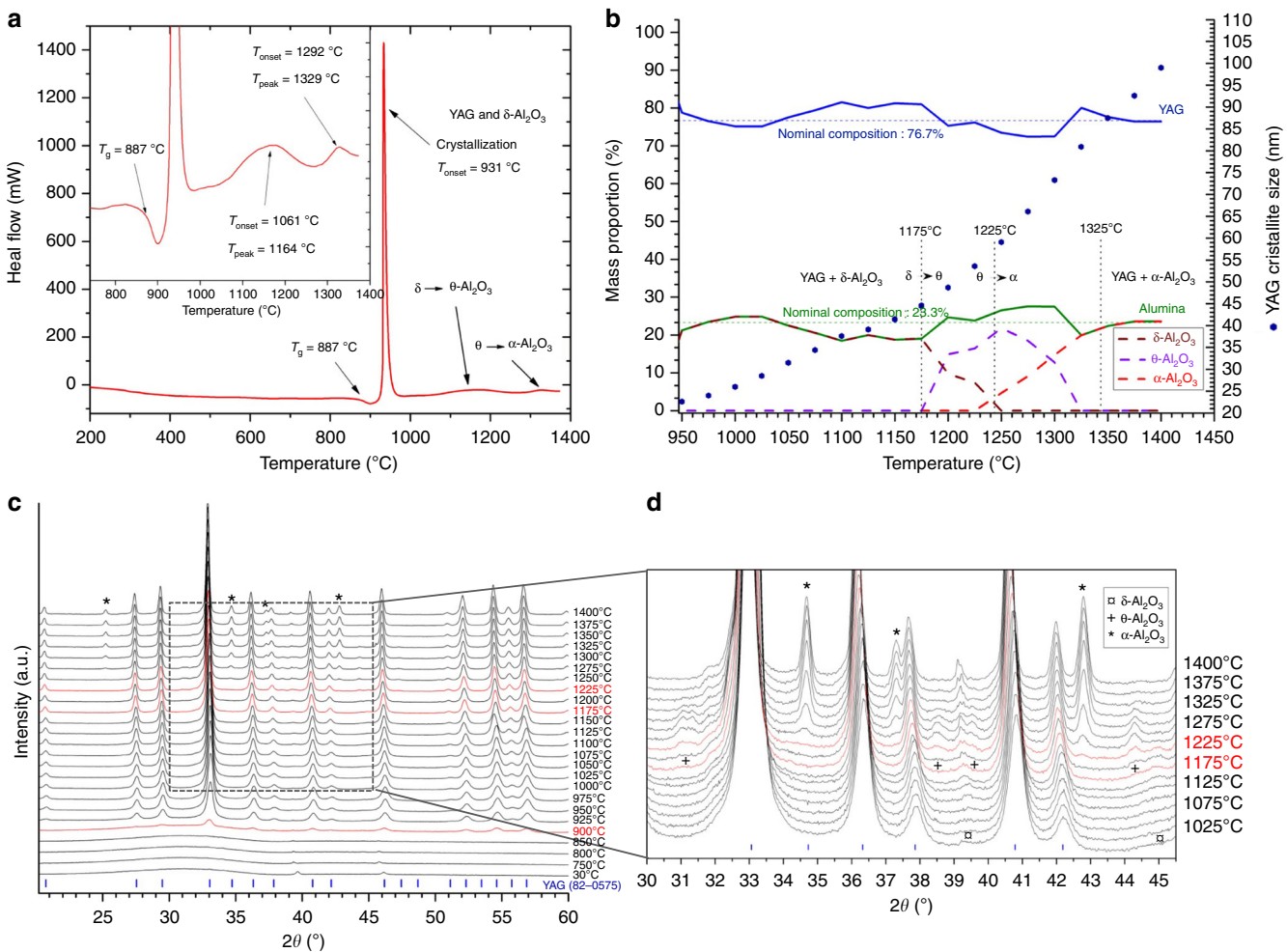

**Fig. 2** Crystallization process of the AY26 glass precursor into YAG-Al$_2$O$_3$ composite nanoceramics. **a** Differential scanning calorimetry measurement of the AY26 glass precursor showing the glass transition at 887±1 °C and a strong exothermic peak corresponding to crystallization of both YAG and δ-Al$_2$O$_3$ phases at 931±1 °C followed by two broad exothermic peaks corresponding to the δ-Al$_2$O$_3$ → θ-Al$_2$O$_3$ ($T_{onset}$ = 1061 °C; $T_{peak}$ = 1164 °C) and θ-Al$_2$O$_3$ → α-Al$_2$O$_3$ ($T_{onset}$ = 1292 °C; $T_{peak}$ = 1329 °C) transitions. **b** Evolution of the YAG and Al$_2$O$_3$ polymorphs contents versus treatment (left vertical axis). The evolution of the crystallite size (right vertical axis) versus temperature is also presented. **c** In situ X-ray powder diffractograms collected upon heating from the AY26 glass powder sample. The indexation corresponds to the YAG garnet structure (PDF 70-7794). **d** Enlargement showing the Al$_2$O$_3$ structural evolution

promising alternative strategy to achieve fully dense polycrystalline materials (i.e., ceramics showing complete absence of porosity)[22–26]. This process, starting from a bulk glass precursor elaborated by fast and cost-efficient glass-forming process, can lead to large scale and shapeable transparent ceramics with high doping concentrations[27,28]. Recently, completely crystallized transparent aluminate ceramics, Sr$_3$Al$_2$O$_6$[23] and BaAl$_4$O$_7$[24], have been successfully prepared using such a bulk glass crystallization route. The reports demonstrate that near-zero volume contraction during glass crystallization is required to avoid crack formation and therefore to obtain a fully dense microstructure[23]. Unfortunately, in addition to the complex elaboration of stoichiometric Y$_3$Al$_5$O$_{12}$ glass bulk[29,30], the large density difference between the glass (~4.08 g/cm$^3$)[31] and the crystalline (4.55 g/cm3) YAG phases prevents transparency to be retained during crystallization using a full and congruent crystallization from glass approach[32]. This is the reason why Tanabe et al. developed transparent YAG glass-ceramics[33,34].

Here we demonstrate the possibility to elaborate transparent YAG-based crystalline materials at room temperature via complete nanocrystallization of a 74 mol% Al$_2$O$_3$–26 mol% Y$_2$O$_3$ (AY26) parent bulk glass. The resulting YAG-Al$_2$O$_3$ composite ceramics

present a fully dense microstructure composed of YAG (77 wt%) and Al$_2$O$_3$ (23 wt%) nanocrystals (Fig. 1). These biphasic ceramics demonstrate transparency from the visible up to the near infrared ranges (6 μm) and improved mechanical properties, especially higher hardness, compared to YAG single crystal and transparent ceramics. When doped by Ce$^{3+}$, the YAG-Al$_2$O$_3$ nanoceramic synthesized at 1100 °C shows a 87.5% quantum efficiency, which is comparable with commercial YAG:Ce$^{3+}$ fluorescent powders (75–90%)[35,36]. The combination of the high-performance mechanical and optical properties with the thermal stability of this YAG-Al$_2$O$_3$ ceramic material, coupled to its facile room pressure preparation, opens great potential applications in wide optical fields such as jewellery, lenses, scintillators and phosphor converters for high-power white-light LED, and laser diodes, which require limited light scattering in the materials[37].

## Results

**Material synthesis and sample preparation.** Transparent YAG-Al$_2$O$_3$ ceramics were elaborated through full crystallization from a transparent yttria-alumina glass which composition deviates from stoichiometric YAG. A controlled Al$_2$O$_3$ excess was indeed

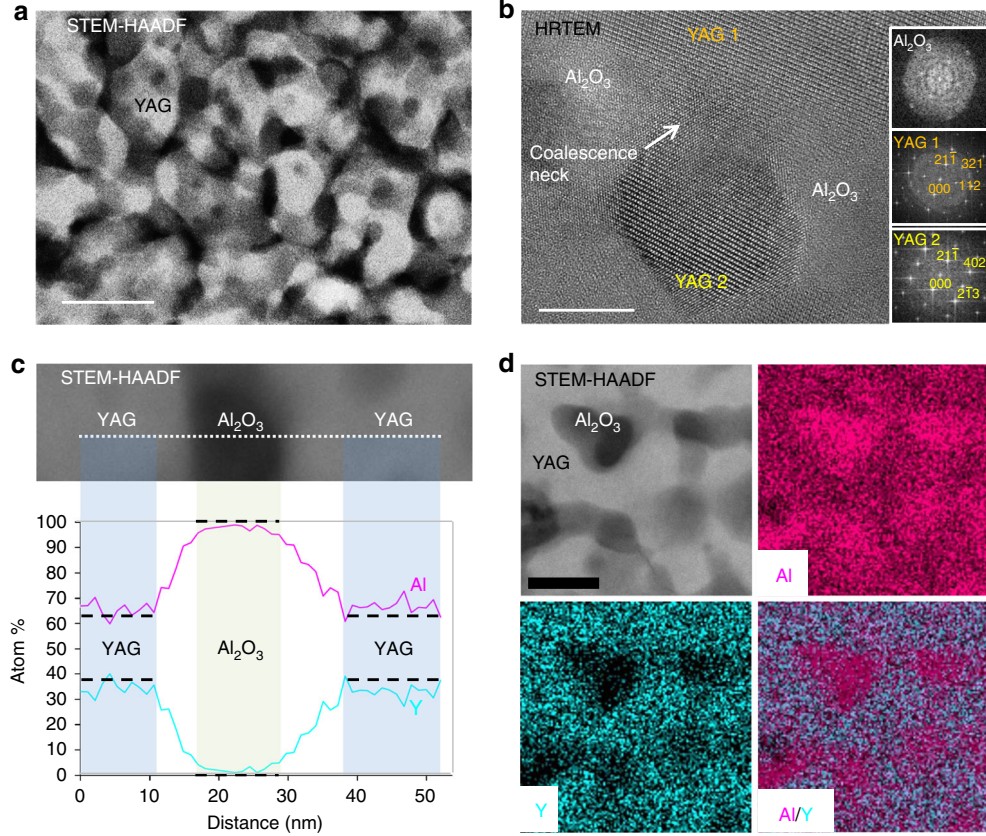

**Fig. 3** TEM study of the YAG-Al₂O₃ ceramic nanostructure. **a** STEM-HAADF image of the YAG-Al₂O₃ ceramic showing an interconnected 3-D network of the YAG crystals (bright phase) separated by Al₂O₃ areas (dark phase). The scale bar corresponds to 50 nm. **b** HRTEM micrograph of the same YAG-Al₂O₃ ceramic showing a typical coalescence neck between two particles (YAG 1 and YAG 2) oriented along the $[3\bar{5}1]$ and the $[\bar{1}42]$ zones axes respectively (scale bar corresponds to 10 nm). The micrograph and the FFT patterns embedded show that the $(21\bar{1})$ plans of the two grains are almost parallel (small rotation deviation of 6°). STEM-EDX cationic **c** composition profile and **d** elemental maps showing that the bright phase observed by STEM-HAADF (scale bar corresponds to 25 nm) can be assigned to YAG and the dark phase to Al₂O₃

required to ensure glass formation and further nanometer-scale crystallization which is required to retain glass transparency in the final biphasic ceramic material. During this study, $Y_2O_3$–$Al_2O_3$ glass samples with different compositions have been synthesized (24, 26, 28, 30, 34, 36, and 37.5 mol% of $Y_2O_3$). However, transparent crystalline samples were only obtained for $Y_2O_3$ contents equal or lower than 26 mol% of $Y_2O_3$, i.e., close to the metastable eutectic point located at 23 mol% of $Y_2O_3$ (Supplementary Fig. 1)[38]. Ceramics with higher $Y_2O_3$ contents, therefore with nominal composition closer to YAG but further away from the eutectic point known to induce possible nanostructures[39], were systematically translucent or opaque. This can be explained by the size of both YAG and $Al_2O_3$ domains exceeding 100 nm (Supplementary Fig. 2). As the crystallization of the AY26 glass composition led to materials with the highest transparency and YAG content, the work presented here focuses on this composition.

Transparent bulk glass precursors (Supplementary Fig. 3) were thus synthesized from a 74 mol% $Al_2O_3$–26 mol% $Y_2O_3$ composition using an aerodynamic levitation system equipped with a $CO_2$ laser[40,41]. This contactless method enables high-temperature melting (at around 2000 °C) and free cooling when lasers are switched off (~300 °C/s). It is expected that scaled, commercial production of larger glass samples could be attained using electric arc high-temperature melting industrial process. The amorphous nature of the AY26 glass was confirmed by both X-ray and electron diffraction (Supplementary Figs. 4 and 5). As

demonstrated by scanning transmission electron microscopy, the glass appears homogeneous down to the nanometer scale (Supplementary Fig. 5). Therefore, the 74 mol% $Al_2O_3$–26 mol% $Y_2O_3$ glass does not show any evidence of phase separation, as reported for various $Al_2O_3$–$Y_2O_3$ melt compositions and which could explain the nanometer-scale crystallization[31,42,43].

Differential scanning calorimetry (DSC) measurement on the AY26 glass collected as a function of temperature clearly shows glass transition at 887 ± 1 °C followed by a strong exothermic peak at 931 ± 1 °C. As illustrated by in situ high-temperature X-ray powder diffraction (HT-XRD), this latter corresponds to the concomitant crystallization of glass into $Y_3Al_5O_{12}$ and $\delta$-$Al_2O_3$ phases (Fig. 2). Two small and broad DSC exothermic peaks can also be observed at higher temperatures, corresponding to the $\delta$- to $\theta$-$Al_2O_3$ ($T_{onset} = 1061$ °C; $T_{peak} = 1164$ °C) and $\theta$- to $\alpha$-$Al_2O_3$ ($T_{onset} = 1292$ °C; $T_{peak} = 1329$ °C)-phase transitions, respectively. These results are in agreement with previous studies on transition alumina-phase transformations from boehmite[44]. The enhanced thermal stability of transition $Al_2O_3$ phases in the AY26 ceramics compared to previous works may be attributed to the large specific interfacial areas of the $Al_2O_3$ nanodomains (~145 m²/g), which can have significant influence on the transformation thermodynamics[45,46]. Following these observations, YAG-$Al_2O_3$ biphasic nanoceramics were then simply prepared by crystallization of the AY26 glass via a single thermal treatment of 2 h at temperatures ranging from 950 to 1100 °C. The resulting YAG-$Al_2O_3$ composite ceramics show transparency even though

limited light scattering can be observed in the visible range when the thermal treatment temperature is increasing (Supplementary Fig. 6).

**Material microstructure.** The room temperature X-ray powder diffraction pattern of the AY26 glass sample crystallized at 1100 °C for 2 h can be indexed with two crystalline phases, $Y_3Al_5O_{12}$ and transition $Al_2O_3$ (the determination of the nature of the $Al_2O_3$ polymorph is not straightforward given that both $\gamma$-$Al_2O_3$ and $\delta$-$Al_2O_3$ phases exhibit close diffraction patterns. However, Rietveld refinement showed better agreement factors using the $\gamma$-$Al_2O_3$ structural model). Quantitative phase analysis performed by Rietveld refinement led to $79 \pm 2$ wt% YAG and $21 \pm 2$ wt% $Al_2O_3$, in good agreement with the nominal formula (77 wt% YAG and 23 wt% $Al_2O_3$). Moreover, using the fundamental parameters approach[47], the crystallite size could first be estimated as $35 \pm 2$ nm. However, a strain effect correction had to be taken into account, which markedly improved the fit and increased the crystallite size to $51 \pm 2$ nm. The AY26 material heat treated up to 1400 °C shows continuous grain growth upon heating as demonstrated by in situ HT-XRD (Fig. 2b). Analysis of these XRD data shows that full crystallization of the YAG-$Al_2O_3$ glass composition is achieved from 900 °C (no phase quantification evolution from this temperature (Fig. 2b, c)). One can note that the full crystallization from glass process enables YAG-based crystalline materials with small nanocrystals to be elaborated using an appropriate single heat treatment. Moreover, the growth rate remains relatively slow at high temperature, most probably because of the presence of $Al_2O_3$ barriers, implying that a self-limited growth mechanism must be taking place[48,49].

The bright field transmission electron microscope (TEM) micrograph presented Fig. 1a and the related selected area electron diffraction (SAED) pattern embedded clearly indicate that strong crystallization has occurred in the material. No amorphous area could be detected, supporting a full crystallization from glass process. Moreover, the nanometer scale of the YAG crystals suggested by XRD is confirmed and the size distribution appears relatively narrow (average size of 26 nm, $\sigma = 6.7$ nm). The presence of strain in the sample will be further confirmed by high-resolution scanning transmission electron microscopy (HRSTEM)-high angle annular dark field (HAADF) and the agreement between the crystal size determined by XRD and TEM will be discussed at that time. Even though no phase separation could be detected in the AY26 glass, the nanometer-scale structuration of the related YAG-$Al_2O_3$ composite ceramic (Fig. 1) is typical of a strong volume crystallization mechanism. The ceramic microstructure appears quite similar to the one observed in glass-ceramics elaborated from a nanometer-scale phase separated glass (spinodal decomposition)[50]. We stipulate that the high cooling rate from the melt prevents phase separation to take place during glass forming but we cannot exclude that nanoscale phase separation may occur upon crystallization heat treatment, which would explain the high nucleation rate and the observed nanostructure.

To better characterize the nanoscale microstructure of the transparent YAG-$Al_2O_3$ ceramics, high-resolution transmission electron microscopy (HRTEM) and STEM-HAADF imaging were performed from thin foils of a ceramic sample crystallized at 1100 °C for 2 h (Fig. 3). Once again the samples appear fully dense as no porosity could be detected. The HRTEM image clearly shows the presence of two different crystalline phases (Fig. 1c; Supplementary Fig. 7). The main one, for which FFT (fast Fourier transform) patterns can be indexed with a garnet structure assigned to YAG, shows dark crystallites with uniform size distribution of about 30 nm (Fig. 1a). Unfortunately, the

small size of the residual bright phase located between the YAG grains and the poor related FFT pattern does not allow unambiguous assignment to $Al_2O_3$ (Supplementary Fig. 7). However, as the difference of average atomic number (obtained from the sum of the atomic numbers (Z) of all atoms composing a phase divided by the number of atoms) between $Al_2O_3$ ($Z_{Al2O3} = 10$) and YAG ($Z_{YAG} = 14$) is much different, STEM-HAADF imaging (also called Z-contrast imaging) appears as an efficient imaging mode to distinguish both phases. Indeed, as presented Fig. 3a, STEM-HAADF images exhibit two different phases with strong contrast difference. STEM-EDX elemental maps and cationic composition profile (Fig. 3c, d) both demonstrate that the dark phase can be unambiguously assigned to pure $Al_2O_3$ and the bright phase to YAG. The presence of pure $Al_2O_3$ crystalline phase after glass crystallization is obviously related to the excess $Al_2O_3$ in the AY26 glass compared to the YAG formula. This result demonstrates that all $Y_2O_3$ present in glass has reacted to form YAG nanocrystals, and thus confirms the 77 wt% YAG crystalline fraction content in the YAG-$Al_2O_3$ ceramic previously determined by X-ray powder diffraction, in agreement with a full glass crystallization process into YAG and $Al_2O_3$.

The homogeneous distribution of thin $Al_2O_3$ areas around YAG nanograins has a great role in limiting their coarsening upon heating. XRD data recorded versus temperature show a slow and progressive increase of the YAG nanocrystals size whereas the YAG content remains constant. Moreover, the HRTEM and STEM-HAADF images (Figs. 1c and 3a, respectively) clearly show interconnected YAG nanocrystals sharing grain boundaries and forming a 3-D network. This microstructure is typical of a coalescence growth mechanism[51,52]. This coalescence effect is clearly illustrated on Fig. 3b, where the misfit between the two YAG grains crystallographic orientations is very small (about 6° between the (21-1) plans). Although this coalescence mechanism takes place, it induces strain at the grain boundaries (coalescence necks), in good agreement with the XRD Rietveld refinement presented Fig. 1b, which required strain correction. The small crystallite size determined by TEM (about 30 nm) is consistent with the average size determined by Rietveld refinement without the use of strain constraint. Now considering the grains as merged, the size of the coalesced domain is much larger, here again in good agreement with the size determined by Rietveld refinement with the use of strain constraint (about 50 nm).

The density of the AY26 glass material increases significantly (from 3.80 to 4.25 g/cm³) during crystallization, implying that large volume shrinkage (11.8%) occurs (Supplementary Fig. 8). Nevertheless, the AY26 ceramic remains fully dense without any porosity or crack. It is reasonable to infer that the important shrinkage stress is effectively released by structural relaxation of the very small transition $Al_2O_3$ domains.

Although the refractive index of $Al_2O_3$ (1.60) does not match that of YAG (1.82)[53], the sample nanostructure enables high transparency to be retained during glass crystallization (the nanosize of both YAG and $Al_2O_3$ grains minimizes light scattering according to the Rayleigh–Gans–Debye theory)[28]. The transmittance spectrum recorded in the UV–VIS–NIR and MIR regions is presented Fig. 1d. The photograph of the sample placed 2 cm above the text demonstrates the good transparency of the YAG-$Al_2O_3$ composite ceramic. The transmittance of the YAG-$Al_2O_3$ ceramic is also compared to the AY26 parent bulk glass, the YAG single crystal and the commercial YAG transparent ceramic (Supplementary Fig. 9). The transparency covers a wide wavelength range from the visible up to infrared regions (6 μm), similarly to YAG single crystals. The absorption band located around 3 μm is attributed to the absorption of the free hydroxyl (OH) group, which is commonly observed in oxide

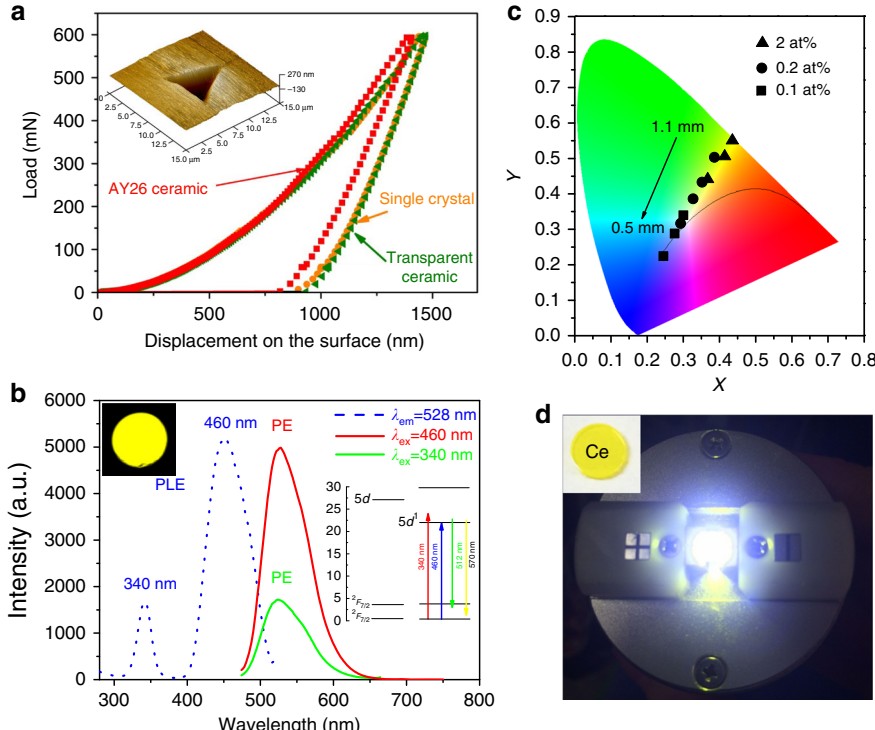

**Fig. 4** Mechanical and optical properties of the YAG-Al$_2$O$_3$ nanoceramic. **a** Typical load-displacement curves of the AY26 ceramic synthesized at 1100 °C and compared to commercial micrometer scale YAG transparent ceramic and YAG single crystal. The indentation morphology of the AY26 ceramic is embedded. **b** Photoluminescence excitation and emission spectra of the Ce$^{3+}$-doped YAG-Al$_2$O$_3$ ceramic. The dotted line is the excitation spectrum; the red and green lines are the photoluminescence emission spectra under excitation at 460 and 340 nm, respectively. The photograph of the ceramic sample recorded under 460 nm laser excitation (4 mW power) and its energy levels diagram are embedded. **c** Color coordinates of the YAG-Al$_2$O$_3$ ceramic under 460 nm LED excitation for various thicknesses (from 0.5 to 1.1 mm) and Ce$^{3+}$ doping contents (from 0.1 at% to 2 at%). The curved line corresponds to the Planckian black-body locus. **d** Photograph of the YAG-Al$_2$O$_3$ ceramic-based WLED under a driving current of 60 mA. The photograph of the ceramic sample (0.1 at% Ce$^{3+}$) is embedded

glasses[54]. However, limited light scattering occurs in the visible range, which can be linked to the presence of a very few YAG crystals larger than the ≈30 nm average size. Moreover, we stipulate that the experimental transmittance measurement is slightly underestimated due to a size/shape effect (incident beam interacting with the edge of the beads) as even the highly transparent AY26 parent glass shows some limited light scattering below 1 μm.

**Mechanical, optical, and thermal properties**. The hardness and elastic modulus of the transparent YAG-Al$_2$O$_3$ composite ceramic crystallized at 1100 °C for 2 h were measured by nano-indentation coupled with an atomic force microscope. YAG commercial transparent ceramic and single crystal were also measured under similar conditions for comparison. Typical load-displacement curves and indentation marks are presented in Fig. 4a. The YAG single crystal and transparent ceramic (of 30 μm average crystallites size, see Supplementary Fig. 10), show almost same hardness and elastic modulus of 21 and 297 GPa, respectively. In comparison, the YAG-Al$_2$O$_3$ nanoceramic presents a higher hardness, 23.6 GPa, whereas the elastic modulus is 281 GPa. The increase of the hardness can be attributed to both the presence of Al$_2$O$_3$ and the nanosize hardening effect of the ceramic which has been described by the classical Hall–Petch relationship[55,56]. These results demonstrate that the transparent YAG-Al$_2$O$_3$ composite ceramic exhibits promising mechanical properties for further applications.

YAG-based ceramics can accept a variety of active doping agents such as Ce$^{3+}$ and Nd$^{3+}$ for fluorescence LED or solid laser

host material applications[1,4,5,37]. Luminescence properties were measured on a Ce$^{3+}$-doped YAG-Al$_2$O$_3$ composite ceramic (AY26 glass crystallized for 2 h at 1100 °C). No segregation of Ce at the grain boundaries could be observed by EDS-STEM experiments even though this point cannot be totally excluded given the low amount of Ce doping. As the ionic radius of Ce$^{3+}$ is very similar to Y$^{3+}$, Ce$^{3+}$ is expected to dope YAG crystals, which provide a good crystal field environment for luminescence. The photoluminescence spectrum presented Fig. 4b shows a typical Ce$^{3+}$: $5d \rightarrow 4f$ broadband emission. The internal quantum efficiency (QE) of the transparent 2% Ce$^{3+}$ substituted YAG-Al$_2$O$_3$ nanoceramic (2% of Y$^{3+}$ ions are substituted by Ce$^{3+}$) reaches 87.5%, similarly to commercial YAG:Ce$^{3+}$ materials, which is promising for further application in the field of white-light LED. The original nanostructure of the biphasic YAG-Al$_2$O$_3$ ceramics may explain the high quantum efficiency by strong confinement effects which induce high luminous efficiency[57,58]. The QE is much superior to that of conventional SiO$_2$-based YAG transparent glass-ceramics (about 30%)[34]. The high QE of the material can also be linked to the high crystalline quality of the YAG nanocrystals in the YAG-Al$_2$O$_3$ composite ceramic[59]. The QE is also higher than the YAG nanoparticles synthesized by soft chemistry[58] (about 54%) which can therefore contain surface defects preventing high QE. The color coordinates of the Ce$^{3+}$-doped YAG-Al$_2$O$_3$ nanoceramic emission under a 465 nm LED excitation show a linear relationship with the thickness and the Ce$^{3+}$ doping concentration (Fig. 4c). Interestingly, the color coordinates for a 0.1% doping concentration and a 1.1 mm thickness are located at ($x = 0.30$ and $y = 0.34$), which

approximates the white-light region. The efficiency, color temperature, and color rendering index of this white-light LED under a driving current of 60 mA are 108 lm/W, 7040 K, and 57.8, respectively. In comparison with the commercial fluorescent YAG:$Ce^{3+}$ materials, the present YAG-$Al_2O_3$ transparent ceramics show great advantages such as low optical loss and high thermal stability.

The thermal conductivity of the small YAG-$Al_2O_3$ composite ceramic disks (4 mm of diameter, ~0.5–1 mm thickness) was successfully measured at different temperatures, as well as YAG single crystal and YAG transparent ceramic reference samples with similar size and shape (Supplementary Fig. 11). The measured thermal conductivity at room temperature of the YAG-$Al_2O_3$ composite ceramic is 4.2 W/m/K. For applications such as phosphor converters for high-power white-light LED and laser diodes, the thermal conductivity exhibits higher values than currently used polymer and glass-ceramic materials (Supplementary Fig. 11)[60,61]. In comparison with the commercial YAG single crystal and YAG transparent ceramic materials (9.8 W/m/K and 9.6 W/m/K, respectively), the thermal conductivity of the YAG-$Al_2O_3$ composite nanoceramic is lower owing to the presence of numerous nano-boundaries inducing heavy phonon scattering as commonly observed in biphasic and nanostructured thermo-electric ceramics[62,63]. Nevertheless, the YAG-$Al_2O_3$ nanocomposite ceramic shows a slow thermal conductivity decrease versus temperature, contrary to rapid decrease in the YAG single crystal and YAG ceramic materials (Supplementary Fig. 11). The temperature dependence of the thermal conductivities of the YAG single crystal and YAG ceramic materials roughly obey the 1/T raw, as a result of the dominant phonon-phonon scattering effect. In the YAG-$Al_2O_3$ nanocomposite, phonon scattering by the numerous nano-boundary becomes much stronger and the consequent thermal conductivity becomes much less temperature dependent. (The interfacial thermal resistance is almost tempera-ture independent above room temperature[64]). Even though the interfacial thermal resistance at the nano-boundaries of YAG-$Al_2O_3$ decreases the thermal conductivity, this effect can be considerably compensated by the presence of secondary phase $Al_2O_3$, which has a much higher thermal conductivity (e.g., 10 W/m/K at 500 °C) than YAG[65]. Moreover, it should be noted that at 500 °C, the YAG-$Al_2O_3$ nanoceramic even presents similar thermal conductivity (3.7 W/m/K) as the YAG ceramic (3.6 W/m/K), and remains slightly lower than the YAG single crystal value (4.7 W/m/K). The thermal conductivity of YAG-$Al_2O_3$ nanoceramic at high temperature could be beneficial for future high-power applications which may be foreseen given the high thermal stability of the YAG-$Al_2O_3$ ceramic beads. Therefore, we anticipate that the YAG-$Al_2O_3$ ceramics will drive the develop-ment of technologically relevant optical and photonic materials.

In conclusion, the synthesis of fully dense YAG-$Al_2O_3$ composite nanoceramics was performed by pressureless and complete crystallization from a 74 mol% $Al_2O_3$–26 mol% $Y_2O_3$ glass. The size of both YAG and γ-$Al_2O_3$ nanocrystals is quite homogeneous and can be tailored as a function of the temperature and duration of the single crystallization heat treatment. At high temperature, both in situ XRD and HRSTEM-HAADF evidence crystal growth via coalescence effect. As a result of the nanometer-scale microstructure, these YAG-$Al_2O_3$ ceramics present transparency from the visible up to the infrared (6 μm) region. The nanostructure of the YAG-$Al_2O_3$ composite ceramics also induces strong mechanical properties (281 GPa elastic modulus and 23.6 GPa hardness). Moreover, under $Ce^{3+}$ doping, the internal quantum efficiency, 87.5%, reaches the level of commercial YAG:$Ce^{3+}$ single crystals. Taking into account the reported mechanical and optical properties, the thermal stability of the material and the simple fabrication

process, these YAG-$Al_2O_3$ ceramics are believed to be promising candidates for wide optical applications such as gem stones, lenses, scintillators and phosphor converters for high-power white-light LED and laser diodes.

## Methods

**Glass synthesis and crystallization.** Commercial oxide powders ($Al_2O_3$, $Y_2O_3$ and $CeO_2$, Sinopharm chemical reagent co. ltd, 99.99% purity) were used as raw materials. Prior to synthesis, all precursors were heated at 800 °C in a muffle furnace for at least 2 h to remove adsorbed water. After weighing, the 74 mol% $Al_2O_3$–26 mol% $Y_2O_3$ (AY26) powder mixture was homogeneously mixed using wet ball milling in ethanol, and pressed into pellets. Bulk samples of ~60–200 mg were then levitated using $O_2$ flow and melted by a $CO_2$ laser at ~2000 °C[40,41,66]. The sample was kept in molten state for about 10–20 s to ensure homogeneity. Turning off the laser then induced rapid cooling (~300 °C/s) and led to glass beads with a diameter of ~2–5 mm (Supplementary Fig. 3). The glass beads were sub-sequently polished into disks (~1 mm thickness) and fully crystallized into trans-parent ceramics by a single crystallization heat treatment in an open-air atmosphere muffle furnace using a temperature between 950 and 1100 °C. $Al_2O_3$–$Y_2O_3$ glasses with various compositions ranging from 24 to 37.5 mol% of $Y_2O_3$ (37.5% $Y_2O_3$ corresponds to the stoichiometric YAG) were also synthesized by aerodynamic levitation coupled to laser heating, and further investigated during this work. The 74 mol% $Al_2O_3$–26 mol% $Y_2O_3$ glass composition clearly leads to ceramics with the highest transparency. Appropriate glass crystallization tem-peratures were determined from differential scanning calorimetry (Setaram MULTI HTC 1600 instrument) measurements performed at a heating rate of 10 K/min, using argon as a purging gas and with alumina pans as sample holders.

**Phase identification and microstructure observation.** Laboratory X-ray powder diffraction (XRD) measurements were performed using a Bragg-Brentano D8 Advance Bruker laboratory diffractometer (Cu Ka radiation) equipped with a lynxEye XE detector. Data were collected from 15 to 130° (2θ) at room temperature with a 0.02 step size and an acquisition time of 10 s per step. In situ X-ray powder diffraction measurements were performed on a similar diffractometer equipped with a linear Vantec detector. The AY26 glass powder was then placed on a platinum ribbon in an HTK16 Anton Paar chamber. Diffractograms were collected between 20 and 60° (2θ) with a 0.024° step size from room temperature up to 1400 °C.

Transmission electron microscopy (TEM) was used to observe the nanostructure of glass and ceramic materials. HRTEM, STEM-HAADF imaging, and EDS analysis were performed on a JEOL ARM200F FEG gun microscope fitted with an Oxford SDD X-Max 100 TLE 100 mm$^2$ EDS system and equipped with a spherical aberration corrector on the probe. Both the glass and ceramic (synthesized from glass at 1100 °C for 2 h) samples were prepared by mechanical polishing using a tripod and inlaid diamond discs down to a 50 μm thickness. Observable foils were obtained by subsequent argon ion milling (PIPS). The ceramic sample was specifically prepared for STEM observations. During the mechanical polishing step realized with a tripod and inlaid diamond disc, a tilt was imposed to form a bevel on one side of the sample. Very thin areas were thus obtained, which enabled to minimize the final argon ion milling (PIPS) step.

**Mechanical, optical, and thermal properties measurements.** The transmittance of the glass and ceramic samples were recorded using a UV–VIS–NIR spectro-photometer (PERSEE TU-1901, Beijing, China) in the 190–2500 nm wavelength range. In the 2500–8000 nm range, the transmittance of the samples was deter-mined by a Fourier-transform infrared spectrometer (Shimadzu FTIR 8200, Kyoto, Japan). Hardness and Young's modulus of the samples were measured using a nanoindenter (MTS-XP) with a force and displacement of 600 mN and 1400 nm, respectively. At least eight indentations per sample were carried out using a Berkovich-type diamond indenter. The indentation morphologies were observed with an atomic force microscope (AFM) (Nanoscope III, Digital Instruments, Woodbury, NY). The indenter load and displacement were continuously and simultaneously recorded during loading and unloading in the indentation process. The hardness and Young's modulus were determined from data acquired during loading based on the Oliver–Pharr model[67]. Emission and excitation spectra (280–800 nm) were recorded on a Hitachi F-7000 spectrofluorometer equipped with a Hitachi U-4100 spectrophotometer. Internal quantum efficiency values were measured by a Photal QE-2100 spectrofluorometer. On the basis of this setup, internal QE was calculated by the following equation:
$\eta = \frac{\text{Number of photons emitted}}{\text{Number of photons absorbed}} = \frac{L_{sample}}{E_{reference} - E_{sample}}$, where $\eta$ represents QE, $L_{sample}$ the emission intensity, $E_{reference}$ and $E_{sample}$ the intensities of the excitation light not absorbed by the reference and by the sample, respectively. $TiO_2$ powder (99.99% purity) was used as a standard reference. White LED modules were realized by combining the YAG-$Al_2O_3$ composite nanoceramics with the blue chips. The efficiency, color coordinates, color temperature, and color rendering index of the white LED were measured by a Spectral radiometer (PMS-80, Everfine, Hangzhou, China).

The thermal diffusivity of the YAG-$Al_2O_3$ nanoceramic disks was measured by laser flash analyzer (Netzsch LFA 427, Bavaria, Germany) from 25 to 500 °C under

vacuum conditions. A specific graphite sample holder of 3.5 mm internal diameter was designed for the small disk samples (4 mm of diameter and ~0.5–1 mm thickness). The Cape–Lehman model[68], which includes both phonon and heat radiative contributions, was applied to evaluate the total thermal diffusivity. All experiments were repeated three times. To validate our experiment, and especially to estimate the incertitude related to the effect of the sample size and shape on the test result, the thermal diffusivity of a commercial YAG single crystal and a micron-scale YAG ceramic (30 μm, provided by the Shanghai Institute of Ceramics, Chinese Academy of Sciences (SICCAS) laboratory) were also measured using similar conditions (same approximate sample size and shape, and same sample holder). The heat capacity of the YAG ceramic and the YAG-Al$_2$O$_3$ nanoceramic were measured from room temperature up to 600 °C with a differential scanning calorimeter (DSC-2910, TA Instruments Corporation). The thermal conductivity $\kappa$ was calculated as $\kappa = \rho * \alpha * c$, where $\rho$ is the density, $c$ is the heat capacity, and $\alpha$ is the thermal diffusivity. The density of the YAG single crystal and the YAG transparent were considered as 4.55 g/cm$^3$, and as 4.25 g/cm$^3$ for the YAG-Al$_2$O$_3$ nanoceramic.

**Data availability**. All relevant data supporting the findings of this study are available from the corresponding authors upon request.

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

## Acknowledgements

This work was financially supported by the National Natural Science Foundation of China (Grant Nos. 51471158, 51674232, 51671181, 51590893), Natural Science Foundation of Beijing Municipality (Nos. 2152032), Chinese Academy of Sciences President's International Fellowship Initiative (Grant No. 2017VEA0010), the National Basic Research Program of China (973 Program, No.2014CB643801), the French ANR towards the FOCAL ANR-14-CE07-0002, the Equipex Planex ANR-11-EQPX-36 projects, and the CNRS (PICS No. 07091 project). We thank the ICMN and CME laboratories (Orléans, France) for TEM and PIPS access, respectively, Prof. Liu R.H. (Grirem Advanced Materials Co., Ltd, General Research Institute for Nonferrous Metals) for QE and white LED modules access, and Prof. Y.B. Pan, Prof. J. Li and Prof. S. W. Wang (Shanghai Institute of Ceramics, Chinese Academy of Sciences) for providing a usual commercial YAG transparent ceramic sample. They also specially acknowledge Prof. W. Pan (Tsinghua University), Prof. J.T. Li, Dr. G. He (Technical Institute of Physics and Chemistry, Chinese Academy of Sciences) and Prof. J. D. Yu (Shanghai Institute of Ceramics, Chinese Academy of Sciences) for helpful discussion and suggestions.

## Author contributions

J.L. and M.A. conceived and designed the project. X.M. synthesized the materials and carried out the characterization with the help from X.L., B.M., E.V., and A.E. All authors participated in measurements and analysis of the data, discussed the results, and took part in producing the manuscript.

## Additional information

**Competing interests:** The authors declare no competing interests.

