## [Peer Review File(PDF 302 kb) · Nature Communications]

Reviewers' Comments:

Reviewer #1:

Remarks to the Author:

The manuscript „Transparent YAG-based nanoceramics via pressure less glass crystallization“ presents new and original investigations of crystallization of glass beads during annealing in air between 950 and 1100°C, formed from a mixture of Al₂O₃, Y₂O₃ and CeO₂ by laser melting at 2000°C and rapid cooling. The microstructure and crystallization had been analyzed by DTA, TEM, XRD, EDS and the materials had been carefully investigated by spectroscopy, hardness measurement and fluorescence spectroscopy. The experimental observations and measurements were discussed in detail.

The complete pressure less synthesis method described in the manuscript shows a clear advantage compared to high pressure processing. However, synthesis demands high temperature and equipment for rapid cooling. Moreover, it is well known that highly transparent YAG ceramics with large grain size can be obtained by fully pressure less slip casting and vacuum sintering at about 1800°C. (see A. Ikesue, Y.L. Aung, V. Lupei, Ceramic Lasers, Cambridge University Press, 2013, ISBN-13: 9780521114080).

The manuscript could greatly be improved if the authors would discuss their findings in relation to these existing ceramics.

1. Fig 1d shows a high transmission in the IR but a clear decrease of transmission in the near-IR and visible range of light. Nanograins of different index of refraction (YAG and Al₂O₃) contribute to scattering of light in the visible range of light. This scattering depends on crystal size of both components and makes materials opaque with increasing grain size. This effect can be estimated by Mie's theory. However, in contrast to single crystals or monophasic ceramics, scattering of light will always be present in such biphasic ceramics.

2. Line 123 and Figure S5 show a decrease of transmission in the visible range with increasing temperature (same effect)

3. Fig 1d: index of refraction and transmission (T_{max}) depend on the wavelength; according to David E. Zelmon, David L. Small, and Ralph Page, "Refractive-index measurements of undoped yttrium aluminum garnet from 0.4 to 5.0 μm," Appl. Opt. 37, 4933-4935 (1998) for YAG results: n(400nm)=1.860 and n(5 μm)=1.722

4. Line 450 (capture of Fig 1d) the data of index of refraction of YAG single crystal and YAG-Al₂O₃ ceramic are reversed

5. Line 82 the hardness of YAG ceramics depend on grain size and may differ from hardness of YAG single crystals

6. Line 84 and 228: fluorescent powders usually have grain sizes of several microns and for Nano powders quantum efficiency decrease because of surface effects. How do the authors explain the high quantum efficiency?

7. Line 97: for the influence of composition (Y₂O₃-Al₂O₃) I would suggest to consider the phase diagram given in: J. L. Caslavsky and D. J. Viechnicki, J. Mater. Sci., 15 [7] 1709-1718 (1980) which shows a metastable eutectic point at 23 mol % Y₂O₃

8. Line 213 and 219: because of the Hall-Petch relationship the grain size of the commercial transparent ceramics should be given for the measurement of the hardness

9. Line 205: similar to the comparison of the hardness a comparison of the transmittance in the visible range of new nanoceramics, single crystal, commercial transparent ceramics would show the differences between the materials

All together the authors present reasonable data and explanations of the effects observed in their experiments. The cited literature is sufficient and the methods were described carefully.

Reviewer #2:

Remarks to the Author:

This paper reported the synthesis of transparent glass-ceramics where YAG nanocrystals were precipitated. The big work includes HT-XRD, HR-TEM, STEM-HAADF/STEM-EDX analysis and FFT pattern analysis of TEM images as well as thermal conductivity and PL quantum efficiency. They successfully obtained transparent glass-ceramics. The picture shown in Fig.1(d) supports its transparency but the decrease in transmission started from 1000 nm and so it can be noticed that the light scattering is significant due to refractive index change between residual Al₂O₃ and precipitated YAG nanocrystals, although the authors did not comment out the aspect. They also reported thermal conductivity and PL quantum efficiency but they did not clearly show the corresponding experimental data (they just wrote the values of thermal conductivity and PL efficiency). Thus, the reviewer cannot judge whether the values were true or not. As long as the sufficient evidences (the detailed experimental data) for thermal conductivity and PL efficiency are not given, the reviewer cannot recommend the paper for Nature Commun. More comments are given below.

(1) The FFT patterns reported should be compared with simulations given for each of Al₂O₃ and YAG phases.

(2) Ce³⁺ ions doped could be located not only in YAG domain but also in boundary region between Al₂O₃ and YAG crystals. Ce³⁺ ions might be doped in Al₂O₃ phase, too. These would have influence on the internal PL efficiency as well as the PL spectral shape. Such consideration should be included in this paper to know the possibility that this kind of nanocomposite could have a high PL efficiency.

Reviewer #3:

Remarks to the Author:

Manuscript NCOMMS-17-12648-T treats the fabrication and characterization of a YAG (Y₃Al₅O₁₃)/Al₂O₃ biphasic composite achieved by the crystallization of yttrium aluminate glass. The stated interest in such a material is for solid state lasers, phosphors, and scintillators; perhaps principally the former most (lasers) since much discussion is offered with respect to short-comings in conventional single crystalline YAG.

The material system is interesting and experiments well-conducted and articulated, particularly the thermal and x-ray analysis. That said, there are a number of issues that need to be addressed prior to the work being suitable for publication, which should be in a journal of lower impact than Nature Communications.

More specifically:

1. The manuscript's English and grammar need to be improved. There are some sections that read well and others less so.

2. The rhetoric needs to be toned down. For example, the word "remarkably" is used, multiple times, which is inappropriate for a scientific paper (and generally not true in this work). Along these lines, several statements do not, to this Reviewer, seem as "remarkable" as the Author's suggest. For example, in the Abstract: "Remarkably, the hardness of these transparent YAG-Al₂O₃ nano-ceramics is 10 % higher than 32 that of previously reported YAG single crystals" – this is not a fair comparison because of the significant Al₂O₃ content. Since Al₂O₃ is considerably harder than YAG, plus its nanoscale dimensions per Hall-Petch, it is expected – not remarkable – that the

hardness of the composite is greater.

3. The literature survey is incomplete and seems to miss completely to work of Tanabe on YAG glass ceramics. While it is true that those materials are not "fully crystallized," they do represent glasses fabricated and crystallized to yield YAG for many of the same applications that this manuscript targets. The most highly cited paper (about 250 citations) is this one: Nishiura, S., Tanabe, S., Fujioka, K. and Fujimoto, Y., 2011. Properties of transparent Ce: YAG ceramic phosphors for white LED. *Optical Materials*, 33(5), pp.688-691; but a very quick Goggle Scholars search found many others; collectively cited at least 1000 times.

4. Perhaps my biggest issue is that the purpose of this work is to realize "YAG" ceramics from glass because of the purported difficulties growing single crystals and, to some extent, conventional ceramics of YAG as well. While this is a common argument, I'm not sure its accurate as single crystalline YAG is ubiquitous (not to mention that the doping level of Er in Er:YAG is 50% which counters the Author's argument about limited doping capabilities). Konoshima has made extremely good YAG ceramics but they have not really been able to supplant the single crystal market. So how serious really is the need to find processing alternatives to YAG single crystal growth? Compounding this is the fact that, in this work, the Author's employ aerodynamic levitation to fabricate the precursor glasses. This is almost laughable since aerodynamic levitation is really only used to make the most difficult of glasses and only yields very small sample sizes. So if the Authors are interested in a scalable and facile approach to "ceramic" YAG, then this is absolutely not the process. That said, the manuscript's rhetoric aside, the work generally is good so it is my advice to the Author's to rewrite this manuscript focusing just on the science of crystallization and phase formation and not promote it as a prospective new process for YAG-based lasers, etc.

5. Other general comments:

a. The Title and much of the paper describes the material as "YAG-based," which I suppose is true. But with approximately 25% of the composite being Al₂O₃, it really is more appropriate to describe it as a fully crystalline YAG/Al₂O₃ composite.

b. I am extremely skeptical about the composite exhibiting higher thermal conductivity than single crystalline YAG (the thermal conductivity of YAG noted in the Abstract is that for the single crystal) or of phase pure polycrystalline (ceramic) YAG. The secondary Al₂O₃ phase would necessarily enhance phonon scattering and greatly reduce thermal conductivity. Unless measurements are actually made, supported by the theory described briefly in the manuscript, such suppositions as to thermal conductivity should be removed.

c. The Author's note the "high transparency" of the composite but it only really attains an equivalent value to that of the glass at a wavelength of about 2000 nm (Figure S5). Firstly, it is not fair to compare the ceramic to the glass since they have differing refractive indices. It would be better to compare the composite to the transmission of single crystalline YAG. Secondly, there is considerably scattering in the visible and near infrared as indicated by the curvature of the transmission curve at shorter wavelengths. Since the composites are suggested to be fully dense, than this scattering is most likely due to its biphasic nature with large index difference (even if scattering dimensions are small) between the YAG and Al₂O₃. For applications in lasers, even minute reductions in transmission (< 0.1%) can markedly alter slope efficiency; this obviously is not so much an issue for applications as phosphors or scintillators. The Authors need to be more careful in their description of the composite as exhibiting "high transparency."

Answers to the Reviewer #1 comments:

General comment: *The manuscript "Transparent YAG-based nanoceramics via pressure less glass crystallization" presents new and original investigations of crystallization of glass beads during annealing in air between 950 and 1100°C, formed from a mixture of Al₂O₃, Y₂O₃ and CeO₂ by laser melting at 2000°C and rapid cooling. The microstructure and crystallization had been analyzed by DTA, TEM, XRD, EDS and the materials had been carefully investigated by spectroscopy, hardness measurement and fluorescence spectroscopy. The experimental observations and measurements were discussed in detail.*

The complete pressure less synthesis method described in the manuscript shows a clear advantage compared to high pressure processing. However, synthesis demands high temperature and equipment for rapid cooling. Moreover, it is well known that highly transparent YAG ceramics with large grain size can be obtained by fully pressure less slip casting and vacuum sintering at about 1800°C. (see A.Ikesue, Y.L.Aung, V.Lupe, Ceramic Lasers, Cambridge University Press, 2013, ISBN-13: 9780521114080).

The manuscript could greatly be improved if the authors would discuss their findings in relation to these existing ceramics.

All together the authors present reasonable data and explanations of the effects observed in their experiments. The cited literature is sufficient and the methods were described carefully.

Answer: We have recently reported the possibility to synthesize large scale (up to 100g samples were prepared in the lab) and highly transparent aluminosilicate (Al Saghir et al., Chem. Mater., 27, 508, 2015) and tellurite (Bertrand et al., Advanced Optical Materials, 4, 1482, 2016) ceramics by full crystallization from glass using largely available commercial furnaces. The melting temperatures were 1700°C and 900°C for the aluminosilicate and the tellurite materials respectively. These examples demonstrate that the full crystallization from glass process can be performed at various temperatures and cooling rates depending on the composition.

Regarding the aluminate compositions detailed in this work, as stated by the reviewer, high temperature and rapid cooling are required. This is why we used the aerodynamic levitation coupled to laser heating system. Nevertheless it is expected that scaled, commercial production of larger glass samples could be attained using a conventional induction or electric arc high-temperature melting industrial process (such as the ones developed by Saint-Gobain CREE in Avignon, France). Moreover, the aerodynamic levitation coupled to laser heating device can be suitable for mass production. A glass bead can be prepared within a few seconds and the French group at CEMHTI is now collaborating with a European jewelry company to automate this process for industrial production (several hundreds of beads can already be produced per hour using a single laser heating device). The Chinese group at IPE is also developing high temperature and high quenching rate alternative processes to aerodynamic levitation for industrial production.

Ikesue et al. have indeed reported the elaboration of transparent YAG ceramics with large grain size by fully pressure less slip casting and vacuum sintering at about 1800°C. We also previously reported the possibility to synthesize several transparent ceramics with large grain size by full crystallization from glass at room pressure. All these recent studies are promising but do not lead to nanoceramics which are of great interest to tailor optical transparency and improve mechanical strength (substantially enhanced for transparent ceramics with nanocrystals compared to large grain size). As suggested by the reviewer, we now acknowledge Ikesue's excellent work in the manuscript and discuss our findings in relation to his works.

Point 1: *Fig 1d shows a high transmission in the IR but a clear decrease of transmission in the near-IR and visible range of light. Nanograins of different index of refraction (YAG and Al₂O₃) contribute to scattering of light in the visible range of light. This scattering depends on crystal size of both components and makes materials opaque with increasing grain size. This effect can be estimated by Mie's theory. However, in contrast to single crystals or monophasic ceramics, scattering of light will always be present in such biphasic ceramics.*

Answer: We agree that limited light scattering occurs in our material, especially in the visible range as some of the crystallites may be slightly larger than the average 30 nm size. This is clearly stated in the manuscript and can now be clearly seen on the new figure S9 which shows the transmittance of the single crystal YAG, the transparent YAG ceramic (commercial material developed at Shanghai Institute of Ceramics, Chinese Academy of Sciences (SICCAS) laboratory), the 74 mol% Al₂O₃ - 26 mol% Y₂O₃ (AY26) parent bulk glass and the transparent YAG-Al₂O₃ biphasic ceramic on the same plot.

We also agree with the reviewer that single crystals as well as monophasic and isotropic (cubic) dense ceramics (YAG, MgAl₂O₄, ZrO₂...) do not face this light scattering problem (as long as all residual porosity can be removed during sintering). In fact, in agreement with Rayleigh theory, light scattering arises when areas larger than \approx 50-100 nm and with different refractive index values are present across the considered material. This can be the case for biphasic ceramics but also for monophasic birefringent ceramics such as Al₂O₃ for example (Al₂O₃ shows hexagonal symmetry and is therefore birefringent). This is why monophasic Al₂O₃ ceramics must be composed of nanometer scale grains well below 100 nm to exhibit transparency.

The application of light scattering to nanometer scale biphasic materials has been developed by several authors, as for the case of nanometer scale glass-ceramic materials (nanocrystals embedded in a glass matrix). These nanometer scale glass-ceramic materials were even shown to exhibit unexpected high transparency, also named "ultratransparency", which was linked to the nanometer scale structuration (S. Hendy et al., Applied Physics Letters, 2002, 81, 1171-1173; M. Mattarelli et al., Physical Review B, 2010, 82 ; P. A. Tick et al., Optical Materials, 2000, 15, 81-91). Light guiding glass-ceramic optical fibers could also be synthesized, proving the high transparency of these materials. So, even though the presence of more than one phase may seem detrimental, it is still possible to synthesize highly transparent materials by minimizing the light scattering effect (which can almost be neglected) via reducing the crystallite size of the material well below 100 nm.

Last, we would like to draw the attention of the reviewer to the fact that the 74 mol% Al₂O₃ - 26 mol% Y₂O₃ (AY26) parent bulk glass also shows limited scattering in the visible range (figure S6 and S9). We stipulate that a bead size/shape effect (interaction of the beam with the edge of the disk samples) might take place for both the glass and the YAG-Al₂O₃ nanoceramic materials. The size of the beam used for transmittance measurements (a mask with a hole of 3 mm was used) is close to the size of the measured beads and, as already observed for previous measurements (Allix et al., Advanced materials 2011 ; Alahrache et al., Chem Mater 2013), small degradation of transmittance measurements at small wavelengths can be observed. This point is now evoked in the revised manuscript.

Nevertheless, as stated in the manuscript, we are not aiming for laser applications but at applications requiring less transparency perfection such as scintillators, lenses, gem stones and phosphor converters for high-power white-light LED and laser diode (these last application require some limited light scattering for efficiency (Li et al., J Mater Chem C, 2013)).

Point 2: *Line 123 and Figure S5 show a decrease of transmission in the visible range with increasing temperature (same effect)*

Answer: As discussed in the previous comment (point 1), using our very simple one step crystallization heat treatment, we did not avoid that a very limited amount of YAG crystals coalesce and become large enough to provide limited light scattering. In situ XRD experiment shows that the crystal size is continuously increasing as a function of the thermal treatment temperature, in agreement with the effect observed figure S5. We have added comments in the manuscript regarding this point.

Even though it is limited, this light scattering effect would definitely be a problem for laser application but we are aiming for different applications such as laser diode, high-power white-light LED, scintillators, lenses and gem stones. Further improvement of the crystallization process, and especially control of the crystal size, might be possible for applications requiring higher transparency.

Point 3: *Fig 1d: index of refraction and transmission (T_{max}) depend on the wavelength; according to David E. Zelmon, David L. Small, and Ralph Page, "Refractive-index measurements of undoped yttrium aluminum garnet from 0.4 to 5.0 μm ," *Appl. Opt.* 37, 4933-4935 (1998) for YAG results: $n(400\text{nm})=1.860$ and $n(5\ \mu\text{m})=1.722$*

Answer: This is perfectly right and rigorous. We used a constant theoretical 85% maximum transmittance in the original manuscript, corresponding to a 1.77 refractive index measured at 589 nm obtained from the literature on a single crystal, by approximating the refractive index to be constant all over the wavelength range. We thank the referee for the reference to Zelmon et al.'s publication which provides more accurate values. We have modified Fig 1d accordingly and cited the proposed reference.

Point 4: *Line 450 (capture of Fig 1d) the data of index of refraction of YAG single crystal and YAG- Al_2O_3 ceramic are reversed.*

Answer: This is right. Thanks for spotting this typo which has been fixed.

Point 5: *Line 82 the hardness of YAG ceramics depend on grain size and may differ from hardness of YAG single crystals*

Answer: We agree that the hardness of YAG ceramics depends on the grain size. It is common knowledge in ceramics that when the grain size decreases, the hardness increases (Hall–Petch strengthening). This is why there is strong interest in developing nanometer scale crystals in transparent ceramics for armor application, such as Al_2O_3 or MgAl_2O_4 . The YAG ceramic is expected to exhibit higher hardness than the YAG single crystal, especially for very small crystals. However, the transparent YAG ceramic we used here is the commercial material developed at the Shanghai Institute of Ceramics, Chinese Academy of Sciences (SICCAS) laboratory, and exhibits large micrometer scale grains (30 microns of average size). Therefore the hardness value of this transparent YAG ceramic is rather similar to the hardness of YAG single crystal as we verified experimentally.

In order to express more rigorously that the YAG ceramic material we used as a reference has similar hardness with the YAG single crystal, we now precise in the text and the related figure caption that we

used a usual commercial YAG transparent ceramic from SICCAS and we also provide a SEM pattern of the microstructure of this ceramic in the new figure S10.

Point 6: *Line 84 and 228: fluorescent powders usually have grain sizes of several microns and for Nano powders quantum efficiency decrease because of surface effects. How do the authors explain the high quantum efficiency?*

Answer: Given that the ionic radius of Ce^{3+} is very similar to Y^{3+} , most of the Ce^{3+} cations are expected to be located in the YAG crystals, which provide a suitable crystal field environment for luminescence. The original nanostructure of our biphasic YAG- Al_2O_3 ceramic may improve the quantum efficiency by strong confinement effects which induce high luminous efficiency (Bhargava et al., Phys. Rev. Lett. **72**, 416 1994 ; Bhargava et al., J. Lumin. **70**, 85 1996 ; D. Haranath, Appl. Phys. Lett. **89**, 173118 2006), and explain why this composite nanoceramic shows quantum efficiency similar to pure YAG.

We agree that there are examples for which quantum efficiency of fluorescent powder decreases with grain size, mostly because of surface effects. This is the case for nanoparticles synthesized by soft chemistry or usual transparent glass-ceramic materials elaborated at lower temperature and which can therefore exhibit surface defects. However, regarding our new YAG- Al_2O_3 nanoceramic material, we believe that in addition to a possible confinement effect there are not many defects on the YAG nanoparticles surface (please note that the “surface” here is in fact an interface between crystals). It is important to note that we used a high crystallization temperature (1100°C) for the reported material. Interestingly, the QE is increasing with the crystallization temperature. At 1100°C, the crystalline quality of YAG is high as seen on HRTEM images. So there may well be surface defects at lower crystallization temperature (or lower crystalline quality of the nanocrystals) but this is not the case at 1100°C.

As for another example of the same effect, we recently published $Cr^{3+}:ZnGa_2O_4$ nano glass-ceramics which show persistent luminescence effect (Chenu et al., J Mater Chem C, 2014). The intensity of the afterglow is stronger than for nanocrystals synthesized by soft chemistry at 700°C and comparable to classic micrometer scale ceramics synthesized by solid state synthesis (work in progress with intensity quantification measurements being performed at S. Tanabe’s lab in Kyoto, Japan).

Point 7: *Line 97: for the influence of composition (Y2O3-Al2O3) I would suggest to consider the phase diagram given in: J. L. Caslavsky and D. J. Viechnicki, J. Mater. Sci., 15 [7] 1709-1718 (1980) which shows a metastable eutectic point at 23 mol % Y2O3*

Answer: We thank the reviewer for this suggestion. We have added the phase diagram as a supplementary figure to the manuscript (figure S1) and the citation to the proposed reference. Moreover, we now comment on the influence of the composition within the Al_2O_3 - Y_2O_3 system in regards to this phase diagram (compositions ranging from 24 to 37.5 mol% of Y_2O_3 have been investigated – note that the 37.5% Y_2O_3 composition corresponds to pure YAG).

Point 8: *Line 213 and 219: because of the Hall-Petch relationship the grain size of the commercial transparent ceramics should be given for the measurement of the hardness.*

Answer: Thanks again for the rigorous comment on the Hall–Petch strengthening effect. We have observed the grain size of the commercial YAG transparent ceramic sample by SEM. The average grain size is about 30 μm . This observation is now stated and commented in the text and an extra figure has been added in the supplementary materials section (figure S10).

Point 9: *Line 205: similar to the comparison of the hardness a comparison of the transmittance in the visible range of new nanoceramics, single crystal, commercial transparent ceramics would show the differences between the materials.*

Answer: We now present a new figure in the supplementary materials section (figure S9) to compare the transmittance of the YAG single crystal, commercial YAG transparent ceramic (SICCAS) and the YAG- Al_2O_3 nanoceramic. This figure shows that both the AY26 parent bulk glass and the YAG- Al_2O_3 nanoceramic materials exhibit some, although limited, scattering in the visible range. As discussed in point 1, we stipulate that a bead size/shape effect (interaction of the beam with the edge of the disk samples) might take place and therefore lowering the transmittance for small wavelengths even though light scattering from crystals larger than the average 30 nm size is thought to take place in the YAG- Al_2O_3 composite nanoceramic.

Answers to the Reviewer #2 comments:

General comment: *This paper reported the synthesis of transparent glass-ceramics where YAG nanocrystals were precipitated. The big work includes HT-XRD, HR-TEM, STEM-HAADF/STEM-EDX analysis and FFT pattern analysis of TEM images as well as thermal conductivity and PL quantum efficiency. They successfully obtained transparent glass-ceramics. The picture shown in Fig.1(d) supports its transparency but the decrease in transmission started from 1000 nm and so it can be noticed that the light scattering is significant due to refractive index change between residual Al_2O_3 and precipitated YAG nanocrystals, although the authors did not comment out the aspect. They also reported thermal conductivity and PL quantum efficiency but they did not clearly show the corresponding experimental data (they just wrote the values of thermal conductivity and PL efficiency). Thus, the reviewer cannot judge whether the values were true or not. As long as the sufficient evidences (the detailed experimental data) for thermal conductivity and PL efficiency are not given, the reviewer cannot recommend the paper for Nature Commun. More comments are given below.*

Answer: Regarding the transmittance decrease below 1000 nm (also answered in point 1 of reviewer 1), we now more clearly acknowledge that the YAG- Al_2O_3 biphasic ceramic show light scattering in the visible range in the updated version. In order to comment and give more details on this aspect, we also propose, in addition to figure S6, a plot gathering the transmittance of the single crystal YAG, the transparent YAG ceramic (commercial material developed at Shanghai Institute of Ceramics, Chinese Academy of Sciences (SICCAS) laboratory), the AY26 parent bulk glass and the transparent YAG- Al_2O_3 biphasic ceramic (figure S9). The light scattering effect is clearly evidenced on this figure and we agree that it can be related to the presence of some crystallites larger than the average 30 nm size (given the refractive index difference between YAG and Al_2O_3).

As discussed in point 1 of reviewer 1, as the AY26 parent bulk glass also shows limited scattering in the visible range (figure S6 and S9), we also stipulate that a bead size/shape effect might take place and therefore lower the transmittance measurements, especially for small wavelengths (interaction of the incident beam with the edge of the disk samples as the beam and the bead have similar sizes – we

have noted similar behaviors on previous studies using glass beads of several sizes ((Allix et al., *Advanced materials* 2011 ; Alahrache et al., *Chem Mater* 2013)).

Regarding thermal conductivity, we did not report any experimental data in the original submission version. In fact it is very tricky to measure thermal conductivity properties from millimeter scale samples. To the best of our knowledge such measurements have never been reported. In the first submitted version we just stipulated that the thermal conductivity might be high by referring to the work of Li et al. (*J. Mater. Chem. C*, 4, 8648, 2016).

Now, following the reviewers' comments and expectations on thermal conductivity properties we have been developing an in-house sample holder to measure thermal conductivity properties on our small disk samples of 4 mm of diameter and ~ 0.5-1 mm thickness.

The thermal diffusivity of the YAG-Al₂O₃ composite nanoceramics were measured by laser flash analyzer (Netzsch LFA 427, Bavaria, Germany) from 25 to 500°C under vacuum conditions. A specific graphite sample holder of 3.5 mm internal diameter was designed for our small samples. The Cape-Lehman model (Cape, Lehman, Temperature and finite pulse-Time effects in the flash method for measuring thermal diffusivity, *J. Appl. Phys.* 34 (7) 1909 (1963)), which includes both phonon and heat radiative contributions, was applied to evaluate the total thermal diffusivity. All experiments were repeated three times. In order to validate our experiment, and especially to estimate the uncertainty related to the effect of the sample size and shape on the test result, the thermal diffusivity of a commercial YAG single crystal and a micron-scale YAG ceramic were also measured using the same test condition (same approximate sample size and shape, and same sample holder). The heat capacity of YAG and YAG-Al₂O₃ nanoceramic from room temperature to 600°C were measured with a differential scanning calorimeter (DSC-2910, TA Instruments Corporation). The thermal conductivity κ was calculated as $\kappa = \rho * \alpha * c$ where ρ is the density, c is the heat capacity, and α is the thermal diffusivity. The density of YAG single crystal and YAG transparent was considered as 4.55 g/cm³, and as 4.25 g/cm³ for the YAG-Al₂O₃ nanoceramic. The results are presented figure S11.

The thermal conductivity at room temperature of the YAG-Al₂O₃ nanoceramic is 4.2 W.m⁻¹.K⁻¹ whereas the measured values for the commercial YAG single crystal and ceramic materials are 9.8 W.m⁻¹.K⁻¹ and 9.6, respectively. Compared to YAG single crystal and ceramic materials, the YAG-Al₂O₃ nanoceramic presents a lower thermal conductivity. The thermal conductivity of the YAG-Al₂O₃ nanoceramic is also lower than the thermal conductivity measured on YAG-Al₂O₃ micrometer scale ceramics (Li et al., *J. Mater. Chem. C*, 4, 8648, 2016) (18.5 W.m⁻¹.K⁻¹). We stipulate that the nanostructuring of the transparent YAG-Al₂O₃ ceramic, which implies numerous nanometer scale nano-boundaries, is inducing heavy phonon scattering. This behavior is commonly observed in biphasic and nanostructured ceramics developed for thermoelectricity (K. Biswas et al., *Nature*, 489, 414, 2012, Wang et al. *Applied Physics Express*, 3, 031101, 2010). In order to complete this study, we have also measured the thermal conductivity of the AY26 glass and the YAG-Al₂O₃ nanoceramic synthesized at different temperatures. As expected the thermal conductivity increases with the crystallization temperature as the increase of grain size reduces the interface thermal resistance.

Nevertheless, the YAG-Al₂O₃ nanocomposite ceramic shows a slow thermal conductivity decrease versus temperature, contrary to rapid decrease in the YAG single crystal and YAG ceramic materials (new figure S11). The temperature dependence of the thermal conductivities of the YAG single crystal and YAG ceramic materials roughly obey the 1/T law, as a result of the dominant phonon-phonon scattering effect. In the YAG-Al₂O₃ nanocomposite, phonon scattering by the numerous nano-boundary becomes much stronger and the consequent thermal conductivity becomes

much less temperature dependent. (The interfacial thermal resistance is almost temperature independent above room temperature, see Yang et al. *Acta. Mater.* **50**, 2309 2002). Even though the interfacial thermal resistance at the nano-boundaries of YAG-Al₂O₃ decreases the thermal conductivity, this effect can be considerably compensated by the presence of secondary phase Al₂O₃, which has a much higher thermal conductivity (e.g. 10 W.m⁻¹.K⁻¹ at 500 °C) than YAG (Kingery et al., *J. Am. Ceram. Soc.* **38**, 251 1955). Moreover, it should be noted that at 500 °C, the YAG-Al₂O₃ nanoceramic even presents similar thermal conductivity (3.7 W.m⁻¹.K⁻¹) as the YAG ceramic (3.6 W.m⁻¹.K⁻¹), and remains slightly lower than the YAG single crystal value (4.7 W.m⁻¹.K⁻¹). The thermal conductivity of YAG-Al₂O₃ nanoceramic at high temperature could be beneficial for future high power applications which may be foreseen given the high thermal stability of the YAG-Al₂O₃ ceramic beads.

The aimed applications for the YAG-Al₂O₃ nanoceramic are not solid state laser materials but materials for jewelry, lenses, scintillators and phosphor converters for high-power white-light LED and laser diodes. For these last applications, the measured thermal conductivity is much higher than the ones of commercial polymer and glass-ceramic materials currently used in such devices (figure S11). Moreover, applications at high temperature may be foreseen given the high thermal stability of the YAG-Al₂O₃ ceramic beads.

In order to integrate these results, the manuscript has been strongly modified. We thank the reviewers for pressuring to perform these measurements which we believe have not been reported earlier on such small samples.

Regarding photoluminescence measurements, as required by the reviewer we now detail the measurement setup in the experimental section of the manuscript. The internal quantum efficiency values were measured by a Photal QE-2100 spectrofluorometer. Based on this setup, internal QY was calculated by the following equation: $\eta = \frac{\text{number of photons emitted}}{\text{number of photons absorbed}} = \frac{L_{\text{sample}}}{E_{\text{reference}} - E_{\text{sample}}}$ where η represents QY, L_{sample} the emission intensity, $E_{\text{reference}}$ and E_{sample} the intensities of the excitation light not absorbed by the reference and the sample, respectively.

Point 1: *The FFT patterns reported should be compared with simulations given for each of Al₂O₃ and YAG phases.*

Answer: We now propose a new figure which shows ED simulations for the experimental ED patterns presented in the manuscript. In order not to overload the figure in the main text we present this figure as a supplementary material (figure S7).

The exercise is quite straightforward for YAG nanocrystals as the TEM patterns presented in the paper were chosen because the YAG crystals were correctly oriented. Therefore a good agreement between the simulated and the experimental patterns can be clearly appreciated. Regarding Al₂O₃, the exercise is tricky, as the FFT pattern is noisy, partly because the Al₂O₃ nanocrystal is not oriented properly. In fact there is no coherence between the orientation of YAG and Al₂O₃ (they grow independently) therefore there is no reason to see an oriented Al₂O₃ nanocrystal close to a YAG nanocrystal.

We remind here that the assignation of Al₂O₃ to the gamma polymorph relies on the XRD Rietveld refinement. ED indexation is in agreement with this assignation. During the Rietveld refinement, no deviation from the expected structure was observed.

Point 2: *Ce³⁺ ions doped could be located not only in YAG domain but also in boundary region between Al₂O₃ and YAG crystals. Ce³⁺ ions might be doped in Al₂O₃ phase, too. These would have influence on the internal PL efficiency as well as the PL spectral shape. Such consideration should be included in this paper to know the possibility that this kind of nanocomposite could have a high PL efficiency.*

Answer: Given that the ionic radius of Ce³⁺ is very similar to Y³⁺, most of the Ce³⁺ cations are expected to be located in the YAG crystals, which provide a good crystal field environment for luminescence. The original nanostructure of our biphasic YAG-Al₂O₃ ceramics may improve the quantum efficiency by strong confinement effects which induce high luminous efficiency (Bhargava et al., Phys. Rev. Lett. **72**, 416 1994 ; Bhargava et al., J. Lumin. **70**, 85 1996 ; D. Haranath, Appl. Phys. Lett. **89**, 173118 2006), and explain why these ceramics show quantum efficiency similar to pure YAG.

As the ionic radius of Ce³⁺ is very different from that of Al³⁺, the doping content of Ce³⁺ in Al₂O₃ crystals is expected to be extremely small even though it cannot be excluded. Given that the expected doping level is very small, no influence on the internal PL efficiency or on the PL spectral shape is expected, in agreement with our PL observations matching the usual excitation and emission of YAG:Ce.

Regarding the possibility that some Ce³⁺ cations might segregate at the grain boundaries, this is a tricky point to prove, especially for such biphasic nanomaterials. During our HRSTEM study we tried to localize the Ce cations but no obvious evidence was found. The doping content of Ce is low therefore the fact that we could not detect its presence even at the grain boundaries is in agreement with the fact that no strong segregation takes place at the grain boundaries otherwise we would expect to detect it with the size of the probe we used (size probe below 1 nm).

We have taken these remarks into account in the revised version of the manuscript.

Answers to the Reviewer #3 comments:

General comment: *Manuscript NCOMMS-17-12648-T treats the fabrication and characterization of a YAG (Y₃Al₅O₁₃)/Al₂O₃ biphasic composite achieved by the crystallization of yttrium aluminate glass. The stated interest in such a material is for solid state lasers, phosphors, and scintillators; perhaps principally the former most (lasers) since much discussion is offered with respect to shortcomings in conventional single crystalline YAG. The material system is interesting and experiments well-conducted and articulated, particularly the thermal and x-ray analysis. That said, there are a number of issues that need to be addressed prior to the work being suitable for publication, which should be in a journal of lower impact than Nature Communications.*

Answer: We would like to point out that we do not claim solid state laser applications for these transparent YAG-Al₂O₃ composite nanoceramics. Both the transparency and the thermal conductivity properties are not matching the standards for such applications. However we believe that these materials may stimulate some interest for many other applications such as lenses, scintillators and gem stones (see recent paper in Nature communication by T. Irifune et al., Pressure-induced nanocrystallization of silicate garnets from glass, 2016), as well as phosphor converters for laser diodes and high-power white-light LED which require limited light scattering materials. It is possible that the reviewer had the impression that we were selling the material for solid state laser applications as we compared much the YAG-Al₂O₃ nanoceramic with the YAG single crystal and YAG transparent ceramic as they are very famous materials. We tried our best to clarify more the aimed applications

right from the start of the paper and we also now compare the YAG-Al₂O₃ composite nanoceramic with other YAG-based materials (such as resin and glass-ceramic phosphors, see figure S11).

Point 1: *The manuscript's English and grammar need to be improved. There are some sections that read well and others less so.*

Answer: We have done our best to revise the manuscript accordingly to the reviewer suggestion.

Point 2: *The rhetoric needs to be toned down. For example, the word “remarkably” is used, multiple times, which is inappropriate for a scientific paper (and generally not true in this work). Along these lines, several statements do not, to this Reviewer, seem as “remarkable” as the Author's suggest. For example, in the Abstract: “Remarkably, the hardness of these transparent YAG-Al₂O₃ nano-ceramics is 10 % higher than that of previously reported YAG single crystals” – this is not a fair comparison because of the significant Al₂O₃ content. Since Al₂O₃ is considerably harder than YAG, plus its nanoscale dimensions per Hall-Petch, it is expected – not remarkable – that the hardness of the composite is greater.*

Answer: We agree that the paper contains some “hypes”. We have made an effort to clean the revised version.

The enhanced mechanical properties of the YAG-Al₂O₃ ceramic compared to YAG single crystal are indeed expected since Al₂O₃ is considerably harder than YAG, plus the Hall-Petch effect linked to the nanoscale dimensions. Nevertheless, from a general material point of view, this mechanical properties enhancement is of great interest for further application. We have taken into account the remark regarding the wording.

Point 3: *The literature survey is incomplete and seems to miss completely to work of Tanabe on YAG glass ceramics. While it is true that those materials are not “fully crystallized,” they do represent glasses fabricated and crystallized to yield YAG for many of the same applications that this manuscript targets. The most highly cited paper (about 250 citations) is this one: Nishiura, S., Tanabe, S., Fujioka, K. and Fujimoto, Y., 2011. Properties of transparent Ce: YAG ceramic phosphors for white LED. *Optical Materials*, 33(5), pp.688-691; but a very quick Goggle Scholars search found many others; collectively cited at least 1000 times.*

Answer: We are well aware and respectful of S. Tanabe's work and contribution to the field. We initially wished to make a difference between previous glass-ceramic materials and these fully crystallized materials, i.e. ceramics. This is why we did not focus on glass-ceramic literature. However we admit that these references are lacking in the original manuscript. Therefore we have added some of them in the revised version.

Point 4: *Perhaps my biggest issue is that the purpose of this work is to realize “YAG” ceramics from glass because of the purported difficulties growing single crystals and, to some extent, conventional ceramics of YAG as well. While this is a common argument, I'm not sure its accurate as single crystalline YAG is ubiquitous (not to mention that the doping level of Er in Er:YAG is 50% which counters the Author's argument about limited doping capabilities). Konoshima has made extremely good YAG ceramics but they have not really been able to supplant the single crystal market. So how*

serious really is the need to find processing alternatives to YAG single crystal growth? Compounding this is the fact that, in this work, the Author's employ aerodynamic levitation to fabricate the precursor glasses. This is almost laughable since aerodynamic levitation is really only used to make the most difficult of glasses and only yields very small sample sizes. So if the Authors are interested in a scalable and facile approach to "ceramic" YAG, then this is absolutely not the process. That said, the manuscript's rhetoric aside, the work generally is good so it is my advice to the Author's to rewrite this manuscript focusing just on the science of crystallization and phase formation and not promote it as a prospective new process for YAG-based lasers, etc.

Answer: We indeed used common arguments to explain why there is research interest for new transparent ceramics synthesis processes. We agree that high performance YAG single crystals and transparent ceramics can be synthesized but some problems do remain. For example, as stated by manufacturers such as Coorsteck, the lack of reproducibility in transparent ceramic elaboration is the major drawback preventing their (large) commercialization. However, and this is an important point which was not clear enough in the first version of the manuscript, we do not aim for laser ceramic applications for the reported YAG-Al₂O₃ ceramic material. If it is right that the initial idea of this project has been motivated by a broad interest in transparent crystalline materials and therefore laser ceramic materials, we do not aim at replacing YAG single crystals and YAG transparent ceramics with the materials presented here. As developed previously in this document (answers to Reviewer 1), applications requiring less transparency and which can be suitable for millimeter scale samples such as lenses, gem stones, scintillators, and phosphor converters for high-power white-light LED and laser diode (these last application require some limited light scattering for efficiency (Li et al., J Mater Chem C, 2016)) seem much more appropriate.

We would like to state that regarding these applications, the aerodynamic levitation coupled to laser heating device can be suitable for mass production. A glass bead can be prepared within a few seconds and the French group at CEMHTI is now collaborating with a European jewelry company to automate this process for industrial production (several hundreds of beads can already be produced per hour using a single laser heating device). The Chinese group at IPE is also developing alternative high temperature and high quenching rate processes for industrial production.

We have modified the manuscript in order to better define the possible applications of the YAG-Al₂O₃ ceramic material.

Point 5: *Other general comments:*

a. The Title and much of the paper describes the material as "YAG-based," which I suppose is true. But with approximately 25% of the composite being Al₂O₃, it really is more appropriate to describe it as a fully crystalline YAG/Al₂O₃ composite.

b. I am extremely skeptical about the composite exhibiting higher thermal conductivity than single crystalline YAG (the thermal conductivity of YAG noted in the Abstract is that for the single crystal) or of phase pure polycrystalline (ceramic) YAG. The secondary Al₂O₃ phase would necessarily enhance phonon scattering and greatly reduce thermal conductivity. Unless measurements are actually made, supported by the theory described briefly in the manuscript, such suppositions as to thermal conductivity should be removed.

c. The Author's note the "high transparency" of the composite but it only really attains an equivalent value to that of the glass at a wavelength of about 2000 nm (Figure S5). Firstly, it is not fair to compare the ceramic to the glass since they have differing refractive indices. It would be better to compare the composite to the transmission of single crystalline YAG. Secondly, there is considerably

scattering in the visible and near infrared as indicated by the curvature of the transmission curve at shorter wavelengths. Since the composites are suggested to be fully dense, than this scattering is most likely due to its biphasic nature with large index difference (even if scattering dimensions are small) between the YAG and Al₂O₃. For applications in lasers, even minute reductions in transmission (< 0.1%) can markedly alter slope efficiency; this obviously is not so much an issue for applications as phosphors or scintillators. The Authors need to be more careful in their description of the composite as exhibiting “high transparency.”

Answer:

a) We indeed used the “YAG-based” expression, as well as other formula such as “YAG-Al₂O₃ ceramic”, or “YAG-Al₂O₃ biphasic ceramic” as the active optical properties (light emission) are arising from the main phase, YAG, which is doped by Ce³⁺. As suggested, in the new version we have replaced the “YAG-based” expressions by “YAG-Al₂O₃ composite ceramic” which, we agree, is more accurate. Nevertheless, we kept the “YAG-based” expression in the title in order to distinguish this work from the reported directionally solidified YAG-Al₂O₃ eutectic composites (Waku et al., J. Eur. Ceram. Soc. 2000; Lorca, J. L. et al., Prog. Mater. Sci. 2006)), as well as to attract a broad audience.

b). In the initial submitted version we stipulated that the thermal conductivity might be high by referring to the work of Li et al. (J. Mater. Chem. C, 4, 8648, 2016). The authors reported very high values of thermal conductivity for micrometer scale YAG-Al₂O₃ biphasic ceramic and a remarkable thermal stability. Now, following the remarks of Reviewers 2 and 3, we have been developing a sample holder to measure thermal conductivity properties on our small disk samples of ~ 3-4 mm of diameter and ~ 0.5-1 mm thickness (we also measured thermal conductivity on YAG single crystal and ceramic of same size and shape to estimate the measurement error on such small samples – in fact the thermal conductivity measurements are slightly underestimated). This is not an easy task on millimeter scale samples and, to the best of our knowledge, such measurements have never been reported before.

As shown by this experimental work detailed in the response to Reviewer 2 and in the new version of the manuscript, the referee is right. Our experimental values are lower than the ones obtained by Li et al (we measured a 4.2 W.m⁻¹.K⁻¹ value, whereas the measured values for the commercial YAG single crystal and ceramic materials are 9.8 W.m⁻¹.K⁻¹ and 9.6, respectively). In fact, if the micrometer scale YAG-Al₂O₃ biphasic ceramic (SICCAS material) shows higher thermal conductivity than single crystalline YAG or of phase pure polycrystalline YAG, this is not the case for our YAG-Al₂O₃ composite nanoceramics.

We stipulate that the nanostructuring of the transparent YAG-Al₂O₃ composite nanoceramics, which implies numerous nanometer scale nano-boundaries, is inducing heavy phonon scattering. This behavior is commonly observed in biphasic and nanostructured ceramics developed for thermoelectricity (K. Biswas et al., Nature, 489, 414, 2012, Wang et al. Applied Physics Express, 3, 031101, 2010). However, contrary to the thermal conductivity of YAG single crystal and YAG ceramic which decreases sharply with the increase of temperature, the YAG-Al₂O₃ nanoceramic shows a slow decrease. At 500 °C, the YAG-Al₂O₃ nanoceramic even presents similar thermal conductivity (3.7 W.m⁻¹.K⁻¹) as the YAG ceramic (3.6 W.m⁻¹.K⁻¹), which remains slightly lower than for the YAG single crystal (4.7 W.m⁻¹.K⁻¹).

In order to complete this study, we have also measured the thermal conductivity of the AY26 glass and the YAG-Al₂O₃ nanoceramics synthesized at different temperatures (figure S11). As expected the thermal conductivity increases with the crystallization temperature as the increase of grain size reduces the interface thermal resistance.

The aimed applications for the YAG-Al₂O₃ nanoceramic are not solid state laser materials but scintillators, lenses, gem stones, and phosphor converters for high-power white-light LED and laser diodes. For these last applications, the thermal conductivity is much higher than the existing polymer and glass-ceramic materials used in actual devices (figure S11). Moreover, applications at high temperature may be foreseen given the high thermal stability of the YAG-Al₂O₃ ceramic beads.

In order to integrate these results, the manuscript has been strongly modified. We thank the reviewers for pressuring to perform these measurements which we believe have not been reported earlier on such small samples.

c). As discussed in point 1 of reviewer 1, we agree that limited light scattering occurs in our material, especially in the visible range as some of the crystallites may be slightly larger than the average 30 nm size. This is now clearly stated in the revised manuscript and as suggested by the reviewer we now propose a new figure S9 which shows transmittance data of YAG single crystal, transparent YAG ceramic (commercial material developed at Shanghai Institute of Ceramics, Chinese Academy of Sciences (SICCAS) laboratory with 30 μm average grain size), the 74 mol% Al₂O₃ - 26 mol% Y₂O₃ (AY26) parent bulk glass and the transparent YAG-Al₂O₃ biphasic ceramic on a same plot (figure S9).

We agree that the YAG-Al₂O₃ composite ceramic and the YAG-Al₂O₃ glass do not have the same refractive index. However, the refractive index of YAG, for both single crystal and transparent ceramic materials, is not the same as the one of the YAG-Al₂O₃ ceramic. So we reckon that the comparison of all materials together, as now presented figure S9, is the best compromise to evaluate the transparency quality of the YAG-Al₂O₃ ceramic.

Last, we would like to draw the attention of the reviewer to the fact that the 74 mol% Al₂O₃ - 26 mol% Y₂O₃ (AY26) parent bulk glass also shows limited scattering in the visible range (figure S6 and S9). We stipulate that a bead size/shape effect (interaction of the beam with the edge of the disk samples) might take place for both the glass and the YAG-Al₂O₃ nanoceramic materials. The size of the beam used for transmittance measurements (a mask with a hole of 3 mm was used) is close to the size of the measured beads and, as already observed for previous measurements (Allix et al., *Advanced materials* 2011 ; Alahrache et al., *Chem Mater* 2013), small degradation of transmittance measurements at small wavelengths can be observed. This point is now evoked in the revised manuscript.

Nevertheless, as stated in the manuscript, we are not aiming for laser applications but at applications requiring less transparency perfection such as scintillators, lenses, gem stones and phosphor converters for high-power white-light LED and laser diode (these last application require some limited light scattering for efficiency (Li et al., *J Mater Chem C*, 2016)).

Reviewers' Comments:

Reviewer #1:

Remarks to the Author:

In the revised version the manuscript „Transparent YAG-based nanoceramics via pressure less glass crystallization” was clearly improved following the hints of the reviewers. The “Supplementary information” gives new valuable data and additional references have been added and discussed in the manuscript. The results are impressive but the new figure S9 shows that there is still some work to do in the future to improve transmission in the visible range. The authors answered all the questions of the reviewers in detail. Because of the unexpected high quantum efficiency, the reference material used for determination of QE should be given in the methods section ln. 367-368.

I would like to thank the authors and suggest the manuscript for publication in Nature communications.

best regards Jens Klimke

Reviewer #2:

Remarks to the Author:

The manuscript is well revised and the required discussion has been done. The present version of manuscript, thus, can be recommended for publication in Nature Commun.

Reviewer #3:

Remarks to the Author:

The Authors are to be commended on the thoroughness of their replies to all of the Reviewers. In general, their responses and edits are appropriate. That said, YAG (and related compositions) is a well studied system with much work on single crystals, ceramics, glass-ceramics, phosphors, phosphors in glass, phosphors in plastic, etc. And while this work is new, it is the opinion of this Reviewer that it does not rise to the impact of Nature Communications. The work, as revised, is very suitable for a journal such as Optical Materials Express or the Journal of the American Ceramic Society.

Editor, Nature Communication

Dear Dr. Kareh,

We submit here a slightly revised manuscript of our original work entitled 'Transparent YAG-based nanoceramics via pressureless glass crystallization' (NCOMMS-17-12648A) which required very minor revision to address the remaining concern of Reviewer #1. We are grateful to the reviewers for their work. The detailed answer to the reviewer's comment is given below.

Answers to the Reviewer #1 comments:

General comment: *In the revised version the manuscript „Transparent YAG-based nanoceramics via pressure less glass crystallization” was clearly improved following the hints of the reviewers. The “Supplementary information” gives new valuable data and additional references have been added and discussed in the manuscript. The results are impressive but the new figure S9 shows that there is still some work to do in the future to improve transmission in the visible range. The authors answered all the questions of the reviewers in detail. Because of the unexpected high quantum efficiency, the reference material used for determination of QE should be given in the methods section ln. 367-368.*

I would like to thank the authors and suggest the manuscript for publication in Nature communications.

Answer: TiO₂ power (99.99% purity) was used as the standard reference materials for determination of QE. We have added this information in the Method section.